# Seed Morphology, Life Form and Distribution in Three *Bromheadia* Species (Epidendroideae, Orchidaceae)

**Emma Ortúñez** [1,2,*,†] **and Roberto Gamarra** [1,2,†]

1   Departamento de Biología, Universidad Autónoma de Madrid, 28049 Madrid, Spain
2   Centro de Investigación en Biodiversidad y Cambio Global (CIBC-UAM), Universidad Autónoma de Madrid, 28049 Madrid, Spain
\*   Correspondence: emma.ortunez@uam.es
†   These authors contributed equally to this work.

**Abstract:** The seed morphology of three species belonging to the genus *Bromheadia* was analyzed under light and scanning electron microscopy. The seeds of *B. cecieliae* and *B. truncata* were studied for the first time. Differences in the qualitative and quantitative characteristics between the terrestrial *B. finlaysoniana* and the epiphytes *B. cecieliae* and *B. truncata* were observed, which were in concordance with the life form. Due to the variability of the seed shapes, a new methodology is proposed to analyze the distance between the embryo and the testa cells, with the aim of demonstrating the presence of air space within the seed. The method is compared to previous formulae used to measure free air space. Furthermore, a new measurement, the angle in twisted testa cells of epiphytic orchids, is proposed, to evaluate the degree of torsion in medial cells. Although the wide distribution of *B. finlaysoniana* could be related to the great buoyancy of their seeds in contrast to the limited distribution of *B. cecieliae*, we consider that environmental factors are more influential than the buoyancy of seeds when understanding the distribution of these taxa. Future studies on seeds morphology in orchid genera with terrestrial and epiphytic taxa will provide new insights into this research.

**Keywords:** testa cells; anticlinal walls; embryo; terrestrial; epiphyte





## 1. Introduction

Orchid seeds are tiny, comprising a pluricellular embryo enclosed in a thin layer of dead cells, which make up the testa [1]. Testa cells show a set of characteristics that have been observed under light and scanning electron microscopes, providing qualitative and quantitative data used to describe the seeds' morphology [2–6]. Previous studies on seed micromorphology have demonstrated the taxonomic value of several traits [7–9], and emphasize its strong correlation with molecular phylogenies [4,10–12]. In Orchidaceae, seed traits are more conservative than other characteristics [4,5].

Two life forms are dominant in the family Orchidaceae: terrestrial and epiphytic. According to Vij et al. [1], terrestrial taxa are primitive. Seed traits have evolved from terrestrial to epiphytic orchids concerning the cell shape, the seed size, the arrangement of the testa cells, or the internal air space between the embryo and testa. Testa cells have evolved from polygonal (quadrangular to rectangular) in terrestrial orchids to elongated in the epiphytic ones. Seed size has been indicated as being larger in terrestrial species [2,13]; however, Arditti and Ghani [3] rejected this assertion. Parallel testa cells along the longitudinal axis are common in terrestrial orchids and twisted ones have only been observed in epiphytic orchids [1,14]. Free air space is variable amongst orchid seeds [3], even though several studies have suggested that air space is larger in terrestrial species [1,15,16].

Arditti et al. [2] proposed the use of two mathematical formulae to estimate the volume of the seed and embryo, by simulating two cones fused at the base in the case of the seed, and using a prolate spheroid in the embryo. With these data, a formula to estimate the percentage of free air space was also proposed. Seed and embryo volumes

and the percentage of free air space, have been calculated in several orchids [7,12,15,17]. However, Zotz et al. [18] obtained negative results with the use of these formulae due to the variability of the seeds' shape, so they proposed a modification for the embryo volume by simulating two cones fused at the base, as proposed for the seed volume.

Arditti and Ghani [3] suggested that bigger aerial spaces in terrestrial species gives the seeds an advantage by enabling them to be dispersed over long distance due to their buoyancy, thus, epiphytic species with lower percentages of free air space have a smaller capacity for dispersal, as was corroborated in later publications [15,16,19]. Investigations into seed morphology and free air space are related to the habitat; it being important to understand the ecology of taxa and its conservation [20]. Differences among qualitative and quantitative features of the seeds in terrestrial and epiphytic orchids have been reported [1,10,14], even in the same genus.

The genus *Bromheadia* Lindl. belongs to the tribe Vandeae and the subfamily Epidendroideae. It comprises 29 species, distributed from Sri Lanka to Papua New Guinea and Queensland in northern Australia [21]. According to Kruizinga et al. [22], most of the species are epiphytic (rarely terrestrial or lithophytic), but *B. borneensis* J.J.Sm., *B. finlaysoniana* (Lindl.) Miq. and *B. pendek* de Vogel are exclusively terrestrial. In the cladogram of the genus based on the morphological traits of the species, the epiphyte *B. alticola* Ridl. appeared as the oldest taxon, the terrestrial species clumped in the same clade, and the rest of the species, all of them epiphytic, are grouped into a third polytomy [23].

Three species are analyzed in this study: *B. cecieliae* Kruiz., *B. finlaysoniana* and *B. truncata* Seidenf.

*B. cecieliae* is an epiphyte only known from Borneo, and it was indicated with a question mark for the Malay Peninsula [22]. It grows on trunks in lower montane to montane mixed or Dipterocarp forests, at 1200–1700 m [22].

*B. finlaysoniana* is distributed from Southeast Asia to northern Australia. It grows as terrestrial in open habitats, more or less disturbed, on loamy to sandy soils, generally at 0–200 m but can be found up to 1100 m [22,24,25]. Chong et al. [26] considered it an extinct species in Singapore. Brummitt [27] assessed that there were no particular threats associated with this taxon, which is categorized as LC (Least concern).

*B. truncata* is distributed from Thailand to Borneo. It grows as epiphyte in lowland to montane Dipterocarp primary forests, up to 1830 m [22,25]. No information about conservation status has been found in the literature for *B. truncata* and *B. cecieliae*.

Although some traits of the seeds in *B. finlaysoniana* have been described [5,28], there is a lack of information about the seeds in the rest of the genus.

The aim of this study is to compare the seeds of the three *Bromheadia* species mentioned above, and to test if the seed characteristics are in concordance with the life form and the distribution range of each species.

## 2. Materials and Methods

The seeds were carefully removed from mature capsules of specimens housed in the herbarium of the Royal Botanic Gardens at Kew (K). The species studied with the localities, collector and vouchers are given in Table 1. The scientific names and authorities are according to POWO [29]. An average of 30 mature seeds from each sample were analyzed under a light microscope (LM), and 10 mature seeds using scanning electron microscopy (SEM).

**Table 1.** List of species studied with life form (E: epiphyte; T: terrestrial), localities, collector (including collector number) and voucher.

| Species | Life Form | Locality | Collector and Number | Voucher |
|---|---|---|---|---|
| *B. finlaysoniana* | T | Malaysia, Sabah: Distr. Sandakan, Leila Forest Reserve, 17-VIII-1971 | *K. Murch* s.n. | K |
| | | Papua New Guinea: Amanab, W Sepita, disturbed growth near road, semi-shade terrestrial, 3-IV-1928 | *R. Brown 1882* | K000482101 |
| | | Thailand: Sangka, Surin, 300 m, by a stream in open evergreen forest, 15-I-1924 | *A.F.G. Kerr 0129* | K000594041 |
| *B. ceceliae* | E | Malaysia, Sarawak: Batu Lawi, 1050 m, near the river, hill slope, 6-V-2002 | *Y. Mahmud et al. S.88176* | K000718611 |
| *B. truncata* | E | Malaysia: Penang, s.f. | *A.C. Maingay 1680* | K |

For light microscopy observations, the seeds were pre-mounted with PVA (polyvinyl alcohol). The samples for the SEM observations were mounted on SEM stubs and coated with gold in a sputter-coater (SEM Coating System, Bio-Rad SC 502, Contra Costa County, CA, USA). They were examined with a Philips XL30, with a filament voltage of 20 kV at SIDI-UAM (Interdepartmental Service of Investigation, Universidad Autónoma de Madrid).

Quantitative data, such as seed size (length and width), embryo size (length and width), the number of cells along the longitudinal axis, the distances between testa and embryo, the torsion angle of the medial cells with respect to the longitudinal axis, and the seed mass were recorded. In addition, the seed and embryo volume and the percentage of free air space were calculated with the mathematical formulae proposed by Arditti et al. [2].

The distance between both ends of the embryo and the apical and basal poles of the seeds, and between the lateral sides of the embryo and the testa, were measured to verify the presence of air space within the seed (Figure 1a). Furthermore, during our study, we observed variability in the arrangement of the medial cells along the longitudinal axis; measuring the torsion angle of these cells to obtain quantitative data suitable for comparison among the taxa (Figure 1b).

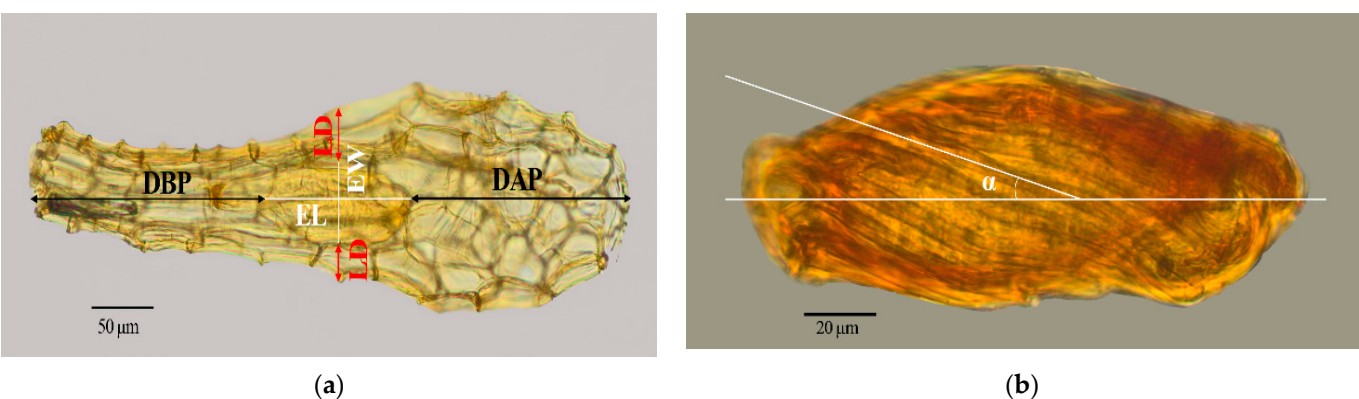

(**a**)  (**b**)

**Figure 1.** Schemes of the methodology followed under light microscopy to measure the values of (**a**) seed of *B. finlaysoniana*, distances between seed and embryo (distance to the apical pole, DAP; distance to the basal pole, DBP; lateral distances, LD; embryo length, EL; embryo width, EW); (**b**) seed of *B. ceceliae*, angle of the medial cells with respect to the longitudinal axis, $\alpha$.

Seed and embryo dimensions were measured under a stereomicroscope Olympus SZ61 using cellSens version 1.4 software (Olympus, Tokyo, Japan).

A sample of 100 seeds for each species were weighted with an analytical balance Mettler Toledo X96 (precision: 0.000001 g) at Chemical Analyses Laboratory (SIDI-UAM); later, the mass of one seed was estimated.

Qualitative data, such as seed shape, morphology and the orientation of medial testa cells, morphology of longitudinal anticlinal walls, periclinal walls (visibility, ornamentation), and the presence of intercellular gaps and waxes were analyzed under SEM, and selected images were recorded.

The terminology and micromorphological methods were adopted from Arditti et al. [2] and Gamarra et al. [14].

Ward's method with Euclidean distances was conducted to obtain a cluster analysis of the quantitative data using PAST software [30].

Distribution maps of each species were created from herbaria collections and bibliographic resources (Table A1) and produced using Map Maker Pro v.3.5 software (Map Maker Limited, 2019) from georeferenced specimens.

The preliminary conservation status was estimated according to the criteria and categories recommended by the IUCN Red List [31], based on the distribution of each taxon.

## 3. Results

### 3.1. Seed Morphology

Seeds of the studied species show differences in the qualitative micromorphological characteristics regarding seed shape, morphology and orientation of medial testa cells, morphology of the longitudinal anticlinal walls, visibility of periclinal walls, and the presence of intercellular gaps and waxes (Table 2). Two different patterns were observed, one in *B. finlaysoniana*, and the other in *B. cecieliae* and *B. truncata*.

**Table 2.** Main micromorphological traits of the seeds in the studied species of *Bromheadia*.

| Taxa | Seed Shape | Medial Cell Shape | Orient. Testa Cells | Long. Anticl. Walls | Pericl. Walls | Intercellular Gaps | Waxes |
|---|---|---|---|---|---|---|---|
| *B. finlaysoniana* | Fusiform to clavate | Rectangular | Parallel | Thin | Visible | Present | Absent |
| *B. cecieliae* | Fusiform | Elongated | Twisted | Thickened | Narrow-to-not visible | Absent | Present |
| *B. truncata* | Fusiform | Elongated | Twisted | Thickened | Narrow-to-not visible | Absent | Present |

*B. finlaysoniana* (Figure 2A) has fusiform to clavate seeds, polygonal and isodiametric cells in the apical pole, rectangular medial and basal cells, medial cells arranged at the same orientation of the longitudinal axis, thin and straight longitudinal anticlinal walls, visible periclinal walls without ornamentation, intercellular gaps at the cell corners (Figure 2B) and a lack of waxes.

*B. cecieliae* and *B. truncata* share the following traits (Figure 2C,D): fusiform seed shape, quadrangular to rectangular cells in both poles, elongated and twisted medial cells along the longitudinal axis, thickened longitudinal anticlinal walls with prominent ridges on both sides of the adhesion zone (anticlinal zone), narrow-to-not visible periclinal walls, cell corners without intercellular gaps, and the presence of waxes in the testa.

The means and standard deviation of length, width and volume of the seed and embryo, percentage of free air space, distance of apical and basal poles between the seed and embryo, distance of the lateral sides between the testa seed and embryo, cell number along the longitudinal axis, the torsion angle of the medial cells with respect to the longitudinal axis, and the estimated mass of one seed for each species are summarized in Table 3.

The data reflect differences mainly in the length and width of the seed, in the distances between the embryo and the seed poles and between the embryo and the lateral sides of the testa in *B. finlaysoniana* with the other taxa (Figure 3).

Seeds in *B. finlaysoniana* are longer and wider than those of *B. cecieliae* and *B. truncata*, and the distances from the seed poles to the embryo and the lateral sides of the testa to the embryo are also larger (Table 3, Figures 3 and 4). The measurements of embryo length and width are similar in the three species (Table 3, Figure 3). The data of the percentage of free air space show negative values for the epiphytic species (Table 3). The number of testa cells along the longitudinal axis is higher in *B. finlaysoniana*. Testa cells are parallelly

arranged to the longitudinal axis in *B. finlaysoniana* but are twisted, in a 17–22° angle in *B. cecieliae* and a 25–29° angle in *B. truncata*. The estimated mass of one seed is higher in *B. finlaysoniana* (Table 3).

The seed size, the distances from the seed poles to the embryo and the lateral sides of the testa to the embryo, the number of testa cells along the longitudinal axis, the angle of the medial cells with respect to the longitudinal axis, and the seed mass, are more similar in *B. cecieliae* and *B. truncata* compared to *B. finlaysoniana* (Table 3).

Using Ward's method with Euclidean distance, two large clusters were produced by the dendrogram (Figure 5), one including the two epiphytic species, and the other with the terrestrial ones.

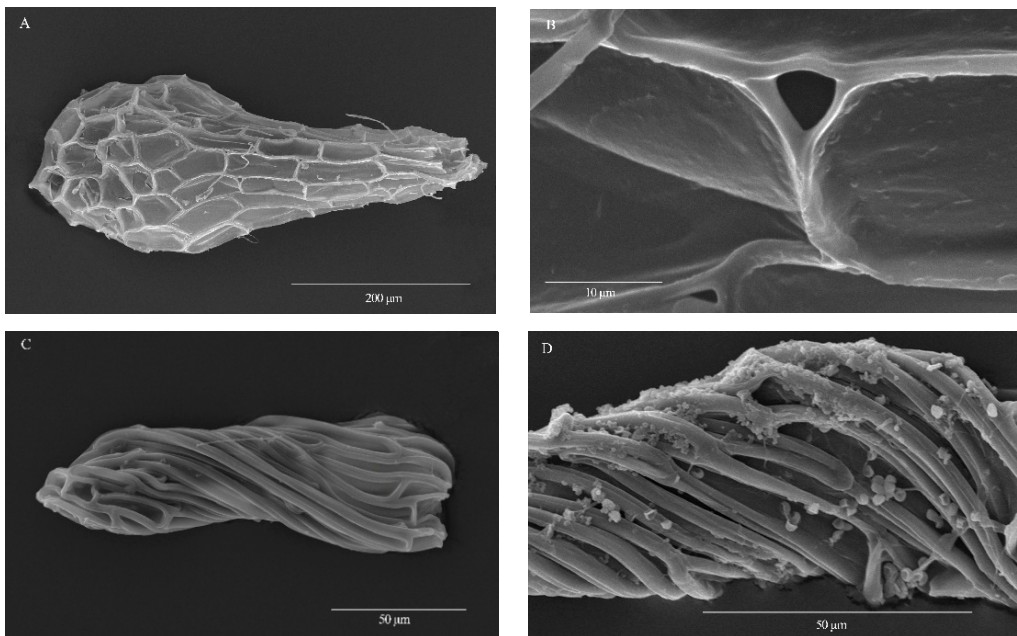

**Figure 2.** Seed morphology under SEM: (**A**) seed and cell shapes in *B. finlaysoniana*; (**B**) intercellular gaps in *B. finlaysoniana*; (**C**) seed and cell shapes in *B. cecieliae*; (**D**) medial cells and anticlinal zone in *B. truncata*.

**Table 3.** Average values and standard deviation of quantitative data in the seeds of the studied species of *Bromheadia*: seed length (SL), seed width (SW), number of cells (NC), medial cell angle (MCA), embryo length (EL), embryo width (EW), seed volume (SV), embryo volume (EV), percentage of free air space, distance to the apical pole (DAP), distance to the basal pole (DBP), lateral distance (LD), seed mass (SM). The seed and embryo volumes, and the percentage of free air space were calculated following the described method [2,3].

| Taxa | SL (µm) ± SD | SW (µm) ± SD | NC | MCA | EL (µm) ± SD | EW (µm) ± SD | SV (mm³ × 10⁻³) ± SD | EV (mm³ × 10⁻³) ± SD | Air Space (%) | DAP (µm) ± SD | DBP (µm) ± SD | LD * (µm) ± SD | SM (µg) |
|---|---|---|---|---|---|---|---|---|---|---|---|---|---|
| *B. finlaysoniana* | 504.41 ± 24.71 | 144.41 ± 9.18 | 7–9 | 0° | 129.70 ± 14.77 | 52.60 ± 4.28 | 2.78 ± 3.94 | 1.56 ± 3.83 | 94.35 ± 1.11 | 175.35 ± 7.26 | 201.14 ± 21.86 | 32.19 ± 6.22 | 0.379 |
| *B. cecieliae* | 148.76 ± 10.36 | 56.25 ± 4.58 | 2–3 | 17–22° | 100.07 ± 9.30 | 48.58 ± 4.31 | 1.24 ± 2.5 | 1.25 ± 2.82 | −0.75 ± 12.21 | 21.43 ± 5.01 | 26.62 ± 8.24 | 3.88 ± 1.31 | 0.120 |
| *B. truncata* | 160.35 ± 13.27 | 51.58 ± 6.30 | 2–3 | 25–29° | 109.32 ± 9.07 | 45.32 ± 6.11 | 1.14 ± 3.15 | 1.14 ± 3.16 | −5.24 ± 8.26 | 20.86 ± 3.00 | 30.16 ± 6.58 | 3.12 ± 0.27 | 0.186 |

\* The lateral distance is the result of the mean of the two values obtained for each measured seed.

The medial cell angle (MCA) is expressed as a range.

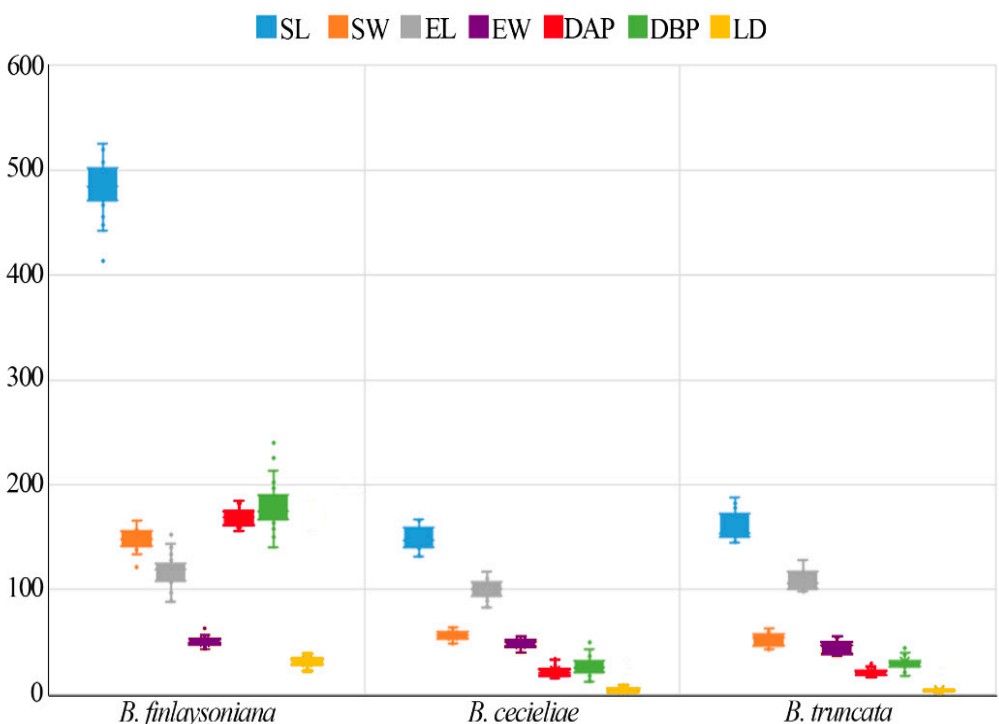

**Figure 3.** Box plots of measured values of the seeds of the three *Bromheadia* species: seed length (SL), seed width (SW), embryo length (EL), embryo width (EW), distance to the apical pole (DAP), distance to the basal pole (DBP), lateral distance (LD).

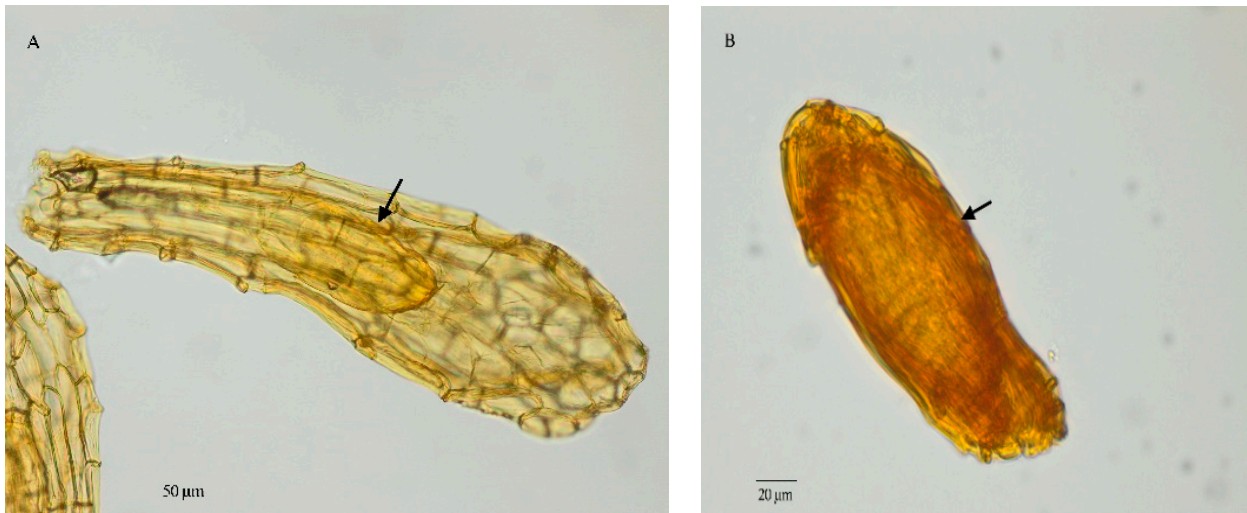

**Figure 4.** Comparison between the seed and embryo (arrow) under LM for (**A**) *Bromheadia finlaysoniana* and (**B**) *B. cecieliae*.

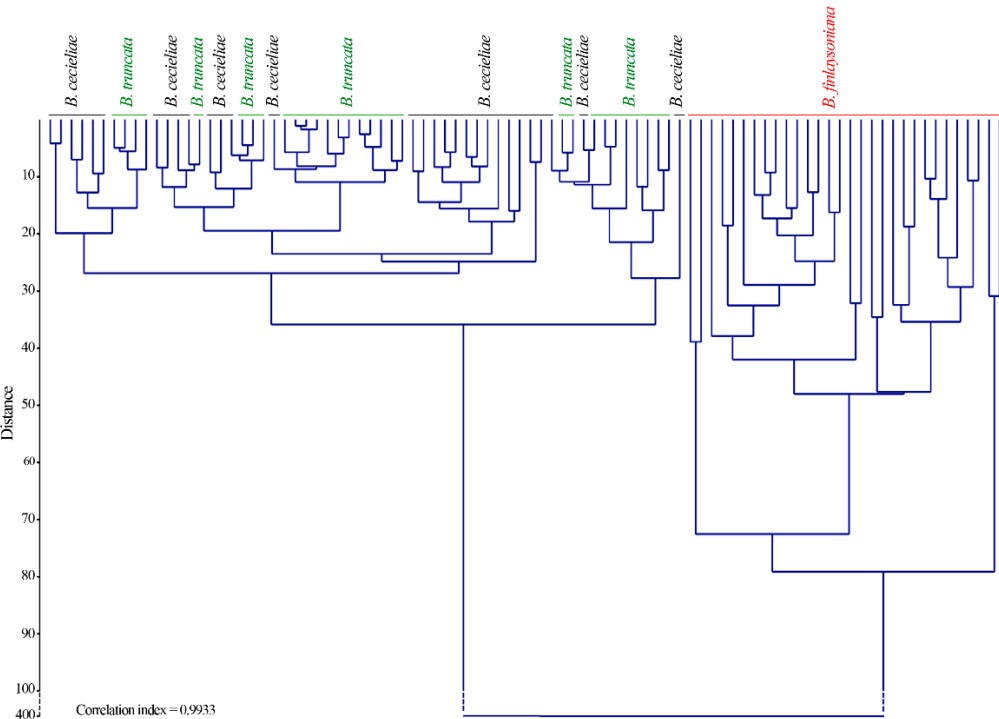

**Figure 5.** Dendrogram showing the clustering of the studied species of *Bromheadia*.

### 3.2. Distribution of the Species

*B. finlaysoniana* is the most widely distributed species within the genus; from Vietnam, Cambodia and Thailand to northern Australia, though mainly in Peninsular Malaysia, Sumatra and Borneo (Sabah, Brunei, Sarawak, Kalimantan) (Figure 6A). It grows preferably in sunny places (rarely semi-shade) as terrestrial in primary and secondary forests (in *Melaleuca* forests with *Pandanus* and *Lophostemon*, Dipterocarp forests, Kerangas forests rich in bryophytes, swamp forests with *Gonystylus*, *Calophyllum* and *Shorea*), heath woodlands, forest edges, scrubby vegetation, natural and artificial grasslands, logging roads, weedy roadside banks, and rocky seashore, in an altitudinal range from 0 to 1100 m, from plains to steep hillsides, in dry soils to waterlogged places, peat swamps, along riversides and streams with Gramineae and Juncaceae, on sandy soils, clay soils, granites or ultramafic substrates, with thick leaf litter (rich in hummus) or on almost bare bedrock.

*B. cecieliae* is endemic from Borneo (Sabah, Sarawak, Kalimantan) (Figure 6B). It grows as epiphyte on tree trunks and twigs, in mossy montane Dipterocarp forests with climbing bamboos and rattans, from 600 to 1900 m. The number of locations is less than 10 in Borneo. Due to its restricted geographic range, a preliminary conservation status of Vulnerable (VU) has been assigned.

*B. truncata* is widespread through Thailand, Peninsular Malaysia, Singapore, Sumatra and Borneo (Sabah), being more abundant in Peninsular Malaysia and Sumatra (Figure 6C). It grows as epiphyte in forests, from 300 to 1370 m. Although the number of locations is low, this species is widely distributed, so its preliminary conservation status as LC (Least concern) is assigned.

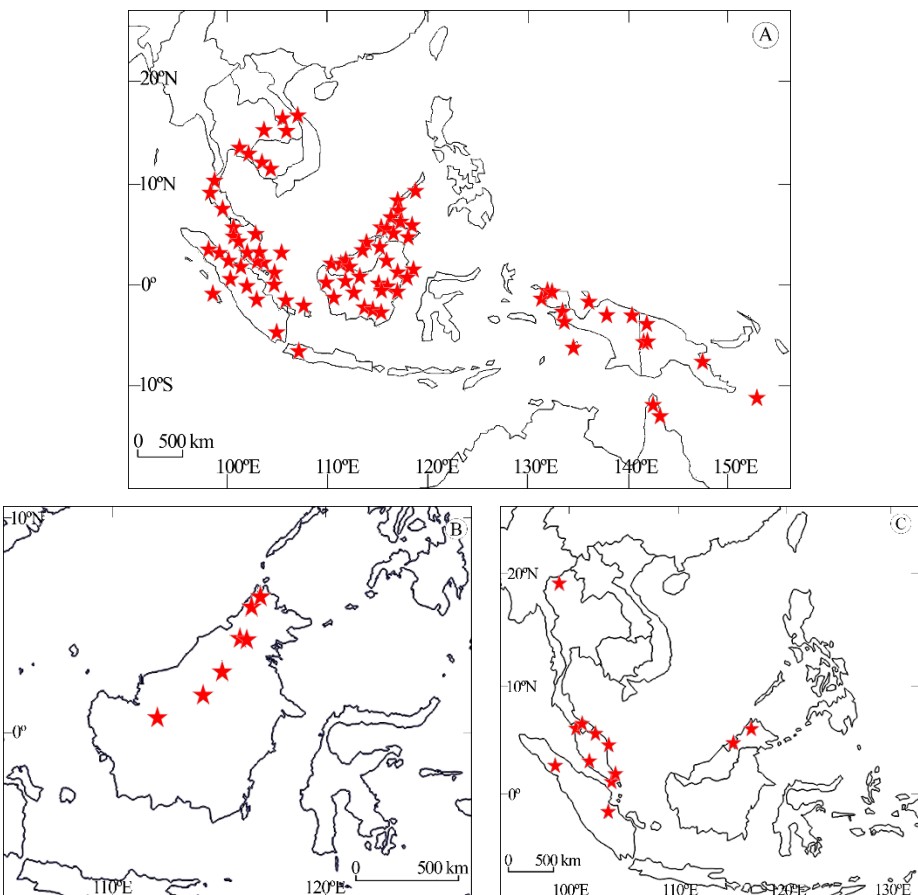

**Figure 6.** Distribution maps: (**A**) *Bromheadia finlaysoniana*; (**B**) *B. cecieliae*; (**C**) *B. truncata*. Red stars represent localities provided in Table A1.

## 4. Discussion

Orchid seeds are extremely small and numerous, and generally dispersed by the air [1,3,5]; however, there is variability between the seeds of different species, even within the same genus [4,5,10,14].

Our results support the data of previous studies acquired by other researchers regarding the qualitative characteristics observed in seeds of terrestrial and epiphytic orchids. In *B. finlaysoniana*, we have observed a set of common traits in the seeds of terrestrial species, such as quadrangular to rectangular testa cells arranged parallel to the longitudinal axis, thin anticlinal walls, straight transversal anticlinal walls, and wide and visible periclinal walls [2,4,5,12,14]. Besides, the presence of intercellular gaps was only found in terrestrial orchids, such as in the tribes Cranichideae and Nervilieae [5,14,32,33]. These results agree with those found in Ziegler [28] and Barthlott et al. [5]. However, in *B. cecieliae* and *B. truncata*, the seeds have long-elongated and twisted testa cells, an anticlinal zone, narrow-to-not visible periclinal walls, and the presence of waxes, a set of common traits in seeds of epiphytic orchids [1,5,14]. These traits have been observed for the first time in the genus *Bromheadia*. Our results verify the presence of qualitative traits regarding the life form, such as in the genus *Liparis* [10,14].

The quantitative data are also in concordance with the life form. Seeds of *B. finlaysoniana* are longer and wider, and the distance from the embryos to both poles and to the lateral sides of the seeds are higher, demonstrating the presence of high internal air space, in concordance with seeds of many terrestrial species [16,17]. In *B. cecieliae* and *B. truncata*, the seeds are shorter and narrower, with the embryos encased in the testa, so the air space is reduced, such as in many epiphytic species [1,10]. The embryo length and width are similar among the three species, only slightly longer in *B. finlaysoniana*. Tsutsumi et al. [10]

asserted that embryo sizes were higher in epiphytic than in terrestrial taxa, but it is necessary to increase the number of samples to corroborate this assertion. Seeds are heavier in *B. finlaysoniana* than in the two epiphytic species. Arditti and Ghani [3] suggested that weight is probably less important than the air space during floatation time.

According to different reports [3,16,19], a higher percentage of free air space in the seed contributes to a higher buoyancy, and the species with this type of seed show wider distributions. In our study, *B. finlaysoniana* is the most widely distributed species within the genus extending from Thailand to northern Australia. It has also been cited from Myanmar [22,34] based on specimens collected by William Griffith. In the label found in the sheet of the New York Botanical Garden (NY02650868), the location is vague ("Birma and Malay Peninsula"). The sheet from the herbarium AMES in Harvard University (HUH023450925) contains two specimens collected in Assam (India) without leaves or flowers, under the name *B. pulchra* Schltr., later revised by H.J. van Scheindelen as *B. finlaysoniana*. Besides, the sheet in the Muséum National d'Histoire Naturelle de Paris (P00436672) contains a label with the vague locality "Indes Orientales", attributed to India in another handwritten label. Kruizinga et al. [22] did not mention this species for India, so its presence in Myanmar and India needs to be confirmed. The wide distribution could be in concordance with the air space in the seed and the terrestrial life form. However, other terrestrial species of the genus show more restricted distributions: *B. borneensis* from Peninsular Malaysia, Lingga Archipelago to Borneo, and *B. pendek* only from Borneo [22]. If the seeds of these species are like those of *B. finlaysoniana*, it would support that the distribution of taxa could be more influenced by abiotic and biotic factors than the buoyancy of seeds.

The vast number of epiphytic species of *Bromheadia* are endemics from Borneo, Papua New Guinea, Peninsular Malaysia, or Sri Lanka [22]. Their distributions would support the lower buoyancy of seeds, such as in *B. cecieliae*, which is native from Borneo. However, *B. truncata* has an extensive distribution with a long distance between the northern locality in Thailand and the central core in the Malay Peninsula and northeastern Borneo (Figure 6C). In this case, the influence of environmental factors would also be more pronounced than the buoyancy of seeds.

The family Orchidaceae developed in their origins as a terrestrial clade, shifted to epiphytic and, later, some lineages returned to the ground [12]. Within the tribe Vandeae, the vast number of species are epiphytic or lithophytic, and fewer than 10 species are exclusively terrestrial [35]. The life form "epiphyte" may be regarded as ancestral in Vandeae, also in the genus *Bromheadia* [23], and the presence of endocarpic trichomes in all the tribe, independently of the life form, supports this assertion [35]. In the case of *B. finlaysoniana*, a regression in the life form to terrestrial has been probably followed with the modification of seed traits such as polygonal cells, thin anticlinal walls, visible periclinal walls, or intercellular gaps, common features in orchid seeds that are more adapted to grow on soil [1,5,14].

The mathematical formulae proposed by Arditti et al. [2] to calculate the seed and embryo volume and the percentage of free air space, and the later modification for the embryo volume by Zotz et al. [18] are not in concordance with the morphological variability of seeds observed in Orchidaceae [1,5,12,14,36]. In our study, the seed shape varies between *B. finlaysoniana* and the two epiphytic species (Figure 2), even if the embryo morphology remains morphologically constant. Using the formulae proposed by Arditti et al. [2], we have obtained negative values for the percentage of free air space. So, we have proposed to measure the distance of the apical and basal poles between the seed and the embryo, and the distances of the lateral sides between the testa and the embryo to check the presence of internal air space. The results of these measurements show that the embryos in the two epiphytic species of *Bromheadia* are encased in the testa because the distances are extremely short; however, in *B. finlaysoniana*, a space is clearly observed between the testa and the embryo.

The reduction in free air space in seeds of epiphytic orchids could be related to less buoyancy [20], a strategy in species that grow on trunks and branches in tropical forests

where the wind speed is lower than in pastures or at the edges of forests. In seeds of terrestrial orchids, a higher buoyancy would contribute to a long-distance dispersal [37]. Kiyohara et al. [38] and Shimizu et al. [39] related buoyancy to a greater length and width of the seeds. In our study, the seeds of the terrestrial *B. finlaysoniana* are longer and wider than those of *B. cecieliae* and *B. truncata*, and the space between the testa and the surface of the embryo is bigger, so a greater buoyancy would contribute to a major distribution [16,19], as shown in Figure 6A. However, the long-distance dispersal of seeds in orchids does not assure germination due to the absence of mycorrhiza or the influence of several abiotic factors [40]. Distribution of orchid taxa is independent of the life form and the seed's buoyancy, as shown by the pantropical distribution of the epiphytic *Polystachya concreta* (Jacq.) Garay and H.R.Sweet [41] or the limited area of the terrestrial *Gymnadenia runei* (Teppner and E.Klein) Ericsson [42].

The conservation status is confirmed for *B. finlaysoniana* [27], due to its widespread distribution and the great variability of habitats and soils in which it grows. For *B. cecieliae* and *B. truncata*, a preliminary conservation status is proposed, although this proposal is limited by the lack of information concerning the ecology of these taxa. Data on characteristics of the phorophytes such as the identification of trees, types of branches, and the height at which the epiphytes grow, are required to assess its conservation status.

Our results have demonstrated that seed morphology differs between epiphytic and terrestrial taxa, indicating that species from the same genus may have different morphologies associated with its life form. In future research, seeds of more *Bromheadia* species will provide new insights. More studies on seed morphology in orchid genera with terrestrial and epiphytic representatives must be encouraged to better understand their ecological processes, orchid evolution, and their adaptations.

**Author Contributions:** Both authors have fully contributed to all stages of the preparation of the manuscript. All authors have read and agreed to the published version of the manuscript.

**Funding:** This research was partially funded by the Departamento de Biología (Universidad Autónoma de Madrid, Spain) under Grant BIOUAM 06-2019.

**Institutional Review Board Statement:** Not applicable.

**Data Availability Statement:** All data generated or analyzed in this study are included in this published article.

**Acknowledgments:** We are much indebted to the curators of the herbarium K for their permission to examine the specimens of the studied genus. The technical assistance of Esperanza Salvador and Isidoro Poveda at the SEM laboratory, and Luis Larumbe at the Chemical Analysis laboratory (SIDI-UAM), is gratefully acknowledged. In addition, we thank Pablo de la Fuente and Guillermo Valdelvira for their comments.

**Conflicts of Interest:** The authors declare no conflict of interest.

# Appendix A

**Table A1.** List of collections and bibliographic resources of the three studied species with countries, localities, collector (including collector number) and voucher. Acronyms of herbaria: BM (The Natural History Museum, London, UK); C (University of Copenhagen, Denmark); CANB (Australian National Herbarium, Canberra, Australia); G (Conservatoire et Jardin Botaniques de la Ville de Genève, Switzerland); HUH (Harvard University, Cambridge, USA); K (Royal Botanic Gardens, Kew, UK); L (Naturalis Biodiversity Center, Leiden, The Netherlands); MO (Missouri Botanical Garden, Saint-Louis, USA); NY (New York Botanical Garden, USA); P (Muséum National d'Histoire Naturelle, Paris, France); PH (Academy of Natural Sciences, Philadelphia, USA).

| Species | Country | Locality | Collector and Number | Voucher |
|---------|---------|----------|---------------------|---------|
| *B. cecieliae* | Indonesia | Borneo, W Kalimantan, Serawai, Sungai Merah | *A.C. Church 2114 et al.* | NY03998648 |
| | Malaysia | Borneo, Mount Kinabalu, Gurulau Spur | *J. and M.S. Clemens 50541* | NY03998647 |
| | Malaysia | Tambunan District, Crocker Range | *J.H. Beaman 10414* et al. | L0283052 |
| | Malaysia | Sabah, S. Rurun headwaters | *J.J. Vermeulen and H. Duistermaet 1067* | L1488365 |
| | Malaysia | Sabah, Long Pa Sia to Long Samadoh | *E.F. de Vogel 8524* et al. | L1488364 |
| | Malaysia | Sarawak, Hose Mountains | *E.F. de Vogel 1174* | L1488361 |
| | Malaysia | Sarawak, Batu Lawi | *Y. Mahmud* et al. *S.88176* | K000718611 |
| | Malaysia | Sarawak, Kelabit Highlands, Bukit Batu Buli | *A. Vogel* et al. | L1488362 |
| *B. finlaysoniana* | Australia | Queensland, Brown Creek, Iron range | *Sine coll.* | CANB |
| | Australia | Queensland, Cape York | *Sine coll.* | CANB |
| | Brunei | Belait Bukit | *M.J.S. Sands 5477* | K |
| | Cambodia | Nord Kampot, Knai | *M. Poilane 14685* | P00460027 |
| | Cambodia | Mulu Prey | *Dr. Normand* | NY03998645 |
| | Indonesia | Lampung, Bangka Island, G. Maras | *A.J. Kostermans 1328 and Anta* | K000482120 |
| | Indonesia | Sumatra, Barat Kota, Siberut Island | *J.J. Smith* | K000482123 |
| | Indonesia | Jambi province, Batanghari Kabupaten Harapan Rainforest | *Wardi* et al. | K000734779 |
| | Indonesia | Sumatra, Langga Pajoeng, Soengei Kanan | *R. Si Toroes 3844* | NY03998654 |
| | Indonesia | Sumatra, Tigapulu Mts, Talang Lakat, Bukit Karampal area | *J.S. Burley 1475a* et al. | MO |
| | Indonesia | Riau Rengat, Tigapulu Mts., Karampal area | *J.S. Burley and Tukirin 1475* | K000482121 |
| | Indonesia | Blitoeng | *Teysmann* | L1488250 |
| | Indonesia | Borneo, Kalimantan, Banjarmasin | *J. Motley 809* | K000482115 |
| | Indonesia | Borneo, Kalimantan, Maruwai | *P.J.A. Kessler 2707* | K000482114 |
| | Indonesia | Borneo, Kalimantan, Haruwu | *J.S. Burley and Tukirin 602* | K000482116 |
| | Indonesia | Borneo, Kalimantan, S. Kahayan, Haruwu | *J.S. Burley 602* et al. | NY00009461 |
| | Indonesia | Moluccas, Aroe Island, Sia | *P. Buwalda 5508* | K000482118 |
| | Indonesia | Irian Jaya, Wasabori, Seroei | *L.J. van Dijk 473* | K000482104 |
| | Indonesia | Irian Jaya, Barat Babo | *Lundquist 648* | K000482102 |
| | Indonesia | Irian Jaya, Barat Fak, Mimika Timur | *E.A. Widjaja 2177* | K000482107 |
| | Indonesia | Irian Jaya, Samberbaba, Seroei | *L.J. van Dijk 829* | K000482106 |
| | Indonesia | Irian Jaya, Cyclops mountains above Hollandia | *C. Koster 4303* | K000482098 |
| | Indonesia | Irian Jaya, Biak Island, Paieri | *A.J. Kostermans 936 and Soegeng* | K000482110 |
| | Indonesia | Irian Jaya, Tablasoefoe | *P. van Royen and H. Sleumer 6446* | K000482095 |
| | Indonesia | Irian Jaya, Barat Fak Fak, Borowai district | *C.J. Stefels 3157* | K000482097 |
| | Laos | Sé-moun | *F.J. Harmand 314* | P00436675 |
| | Malaysia | Sabah, Leila Forest Reserve, Distr. Sandakan | *K. Murch s.n.* | K |
| | Malaysia | Kepong | *Sine coll.* | C |
| | Malaysia | Sarawak, Bako National Park | *J.W. Purseglove 4899* | K000482133 |
| | Malaysia | Sarawak, Sungai Likau, Similajau National Park | *A.B.G. Mohtar and H.J. Othman 59456* | K000482139 |
| | Malaysia | Sarawak, Kuching District, Mount Serapi | *J.H. Beaman 11540* et al. | K000482136 |
| | Malaysia | Sarawak, Serian | *H.J. Othman and A. Munting 61607* | K000482135 |
| | Malaysia | Sarawak, Lundu Kampung Biawak | *A. Munting 56380* | K000482138 |
| | Malaysia | Sarawak, Bario | *P. Sie 35387* | K000482132 |
| | Malaysia | Sarawak, between Bario and Pa Umor | *J.H. Beaman and G. Ismail 11234* | K000482137 |
| | Malaysia | Sarawak, Kelabit Highlands, Kalimantan | *H. Christiansen and F.L. Apu 8* | K000482134 |
| | Malaysia | Perak Larut, Kampong | *Kiah 299* | K000482088 |
| | Malaysia | Johor, Bukit Tinggi | *Sine coll.* | K000482094 |
| | Malaysia | Sabah, Long Pasia | *A. Hoare and L. Beliau 36* | K000482130 |
| | Malaysia | Sabah, Keningau | *S. Sazana* et al. | K000342079 |
| | Malaysia | Anambas Islands, Telok Padang, Jemaja | *M.R. Henderson 20439* | K000482092 |
| | Malaysia | Penang Hill | *A.F.G. Kerr* | K000597047 |
| | Malaysia | Perak Kampong Pokok Assam | *L. Wray 3121* | K000482090 |
| | Malaysia | Pahang Rompin Leban Chondong | *J.H.R. Evans* | K000492089 |

**Table A1.** *Cont.*

| Species | Country | Locality | Collector and Number | Voucher |
|---|---|---|---|---|
| | Malaysia | Sabah, Sandakan | *Keith 6717* | K000482127 |
| | Malaysia | Sabah, Keningau Nabawan Syarikat | *K. Fidilis 128074* | K000482129 |
| | Malaysia | Johor Kampong, Hubong, Endau | *Kadim bin Tassim and Noor 372* | K000482093 |
| | Malaysia | Sabah, Lahad DAtu, Mount Silam | *J.H. Beaman 11621* | K000482131 |
| | Malaysia | Negeri Sembilan, Pulau Rumbia, Sembilan Islands | *C. Boden-Kloss* | K000482086 |
| | Malaysia | Sabah, Sipitang | *A. Cuadra 4063* | K000482128 |
| | Malaysia | Keningau Distr. | *J.J. Wood 753* | K |
| | Malaysia | Perak Larut and Matang Larut Hao | *Sine coll.* | K000482087 |
| | Malaysia | Sabah Beluran, Sg. Tungud | *Sine coll.* | K000482126 |
| | Malaysia | Johore Bahru | *C.W. Franck 284* | P00436678 |
| | Malaysia | Kampong Pulau Domar | *J. Sinclair* | P00436679 |
| | Malaysia | Sabah, Leila Forest Reserve, Distr. Sandakan | *K. Murch* | K |
| | Malaysia | Sarawak, Bako National Park, Telok Pandan | *B.C. Stone 684* | PH00594440 |
| | Malaysia | Sarawak, Marudi District, Bario | *T.E. Beaman 184 and R. Repin* | NY03998652 |
| | Malaysia | Sarawak, Kuching | *M. and J. Clemens 6678* | NY03998655 |
| | Malaysia | Sarawak, Sengghai | *Sine coll.* | NY03998659 |
| | Malaysia | Sarawak, Bako National Park, Telok Asam | *J.W. Purseglove 4899* | NY03998660 |
| | Malaysia | Sarawak, Kuching District, Mount Serapi | *J.H. Beaman 11540* et al. | NY03998653 |
| | Malaysia | Mount Matang | *J. and M. Clemens 22401* | MO |
| | Malaysia | Malacca, Pulau Besar | *B.C. Stone 11494* | PH00594439 |
| | Malaysia | Kedah, G. Jerai | *B.C. Stone 12701a* | PH00594441 |
| | Malaysia | Terengganu, P. Redang | *K.C. Liew 172* | PH00594442 |
| | Malaysia | Sabah, Kota Kinabalu District, Bukit Padang | *J.H. and R.S. Beaman 6836* | MO |
| | Papua New Guinea | Kaiser-Wilhelmsland, Jaduna | *R. Schlechter 19288* | G00165063 |
| | Papua New Guinea | Kiunga subdistrict, Ingembit | *Ridsdale 33240* et al. | K000482100 |
| | Papua New Guinea | Western Amanab, W. Sepik | *R. Brown 1882* | K000482101 |
| | Philippines | Tagalinog Island, Palawan | *D.R. Mendoza and R. Espiritu 91317* | K000482124 |
| | Singapore | Pasir Panjang | *M. Togasi* | K000482091 |
| | Thailand | Ranong prov., Muang Lan | *G. Seidenfaden and Smitinand GT6137* | C |
| | Thailand | Ban Na, Surat | *A.F.G. Kerr 0427* | K000597044 |
| | Thailand | Tako, Langsuan | *A.F.G. Kerr 0380* | K000597039 |
| | Thailand | Bangsak, Trang | *A.F.G. Kerr 0831* | K000597043 |
| | Thailand | Satul | *A.F.G. Kerr 0467* | K000597042 |
| | Thailand | Saba Yoi, Songkla | *A.F.G. Kerr* | K000597045 |
| | Thailand | Kampengpet, Songkla | *A.F.G. Kerr* | K000597048 |
| | Thailand | Sangka, Surin | *A.F.G. Kerr 0129* | K000597049 |
| | Vietnam | Lang-Than | *Thorel* | P00436677 |
| | Vietnam | Quang Nam Prov., Dai Loc Distr., Dai Hong | *L. Averyanov* et al. | P01019665 |
| *B. truncata* | Indonesia | Sumatra: Sicikeh-Cikeh Forest | | Hartini [43] |
| | Indonesia | Sumatra, Jambi | | Hartini [43] |
| | Malaysia | Johor, Gunung Panti area | *Sine coll.* | L1488171 |
| | Malaysia | Terengganu | *G.P. Lewis 100* | K |
| | Malaysia | Selangor | *Segerbäck 2128* | C |
| | Malaysia | Penang | *A.C. Maingay 1680* | K |
| | Malaysia | Borneo, Kinabalu, Gurulau spur | *C.E. Carr* | HUH02341030 |
| | Malaysia | Sabah, Ulu Kalang | *A. Lamb 2004/1116* | L1488170 |
| | Singapore | Chawchu Kang | *H.N. Ridley* | BM000629516 |
| | Thailand | Doi Suthep | *G. Seidenfaden and Smitinand GT2691* | C |
| | Thailand | Waeng Forest Station | *G. Seidenfaden and Smitinand GT7535* | C |

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
