# Peer review of "Seed Morphology, Life Form and Distribution in Three Bromheadia Species (Epidendroideae, Orchidaceae)"

_diversity, doi:10.3390/d15020195_

Round 1
Reviewer 1 Report
The paper describes morphological characteristics of the seeds of three orchid species of the genus Bromheadia with different distribution ranges. The authors propose a new method to characterize the seed morphology of orchid species. The importance of seed morphology studies is emphasized by the attempt to link the distribution range of the species and seed traits related to dispersal.
In the Introduction, it is advisable to explain what are the advantages of using morphological traits to compare different species, in addition to molecular methods that are widely used to construct phylogenies.
The distribution range of the species plays an important role in this study, so the authors should explain has it been it described earlier or is it the first description? How were the herbaria collections and literature sources chosen to ensure that the maps of the distribution areas of the three species are comprehensive?
In the Discussion, the authors write: “During our research on seed orchids, we have found discrepancies in the results of the seed volumes using these formulae due to the variability in seed morphology.” (Ln247) Why are these results (the calculated seed volume) not shown? Comparison of the previously proposed formulae to the method of describing seed morphology proposed by the authors would benefit the paper.
In general, corrections are required to improve the clarity of the text and a grammar check should be performed.
Suggestions and questions are listed below. Please note that more grammatical corrections might be required, it is advisable let the text be corrected by a native speaker.
Ln16 discussed with previous formulae – compared to previous formulae
Ln17 “it is proposed as new the measurement of the angle in twisted testa cells” - the meaning is not clear, perhaps change to: “a new measurement, the angle in twisted testa cells, is proposed [to characterize seed shape? Surface?]”
Ln 27 “Testa cells showed” - show
Ln29 data – data that can be used to describe/characterize these seeds?
Ln31-32 “the evolutionary trends in the seeds have derived in changes” - is "derived" the correct verb? The meaning appears to be that the evolutionary trends were expressed in the changes (…). Maybe I misunderstood the meaning of this sentence. Could it mean that some seed characteristics appeared in the more evolutionary young life form as derived traits?
“Testa” and “seed coat” are synonyms, and only one term should be used throughout the text.
Ln34 – in epiphytic orchids (or “in the epiphytic ones”)
Ln40 who proposed a formula?
Ln46 “that bigger aerial spaces in terrestrial species have advantages” – gives [the seeds] an advantage by enabling them to be dispersed at long distance …
Ln50 “qualitative and quantitative data” - characteristics? Features?
Ln56 – it would be good to mention that the cladogram was based on morphological traits/characteristics of the plants (not a phylogeny based on molecular markers, for example)
Ln60-61 – the meaning of the sentence is clear, but it could be re-written. E.g. “While some traits of the seeds in Bromheadia finlaysoniana have been described 60 [5,19], there is lack of information about the rest of the genus.”
Ln62 “to check the differences between” - rather, to compare the seeds (or seed characteristics) of one terrestrial 62 species and two epiphytic species.
Ln63 “to verify” - to test?
Ln64-65 “... if the distribution range could be in concordance with the variability of seeds and the life form of the studied species.” - this part of the sentence is unclear. It appears that the aim was to test, whether certain seed traits (characteristics) and the life form of the orchids correspond to the species distribution range or can help predict this range. Please clarify.
Ln67 seeds (plural)
Table 1 should be placed closer to the place there it is first mentioned in the text
Ln70 using light microscopy (or under a light microscope)?
Ln79 “the torsion angle of the medial cells respect to the longitudinal axis” – with respect to the axis
Figure 1: are a and b seeds of two different species? The species should be specified in the caption.
Ln96 the weight of one seed (or better – the mass of one seed).
I suggest to use “seed mass” instead of “seed weight”. Later in the text (Table 3) the same abbreviation “SW” is used for two different values (seed width and seed weight).
The seeds of orchids are very small and the mass of 100 seeds would be very small. What was the precision of the balance used?
Ln124 has fusiform to clavate seeds (not “shows”)
Figure 2: presumably, these are SEM images, but it should be specified in the caption.
Ln139 “quantitative data show” Suggestion: “The means and standard deviation of the quantitative characteristics [list] are summarized (or shown) in Table 3.”
The weight of 100 seeds (+/- error) can be reported directly.
Ln144 “mainly in length and width seed and embryo” - this is unclear. Differences in the length and width of the seed and of the embryo? The same question about the distance.
In my opinion, Figure 3 is unnecessary, because all the information is already given in Table 3.
Ln166 “, the distances of the poles and lateral sides between testa seed and embryo" - this part of the sentence is unclear. “the distances between the embryo and the seed poles and the embryo and the lateral sides of the testa”?
Ln169 more similar – compared to B. finlaysoniana?
Ln171 “and the most widely” - and it is the most widely...
Ln172 – its distribution area ranging from
The sub-section 3.2 (Distribution of the species) is very short and is not integrated into the rest of the text.
Ln180 orchid seeds are tiny (not seed orchids)
The sentence should be corrected. Perhaps, it can be emphasized that seeds of all orchids share some common morphological characteristics (small size, reduced nutritional tissues) and dispersal traits (are wind-dispersed), but still there is variability between seeds of different species.
Ln182 precedent – results/data of previous studies or acquired by other researchers?
188 I am not sure that “concordance” is the appropriate word. In accordance with?
Ln211 How was it established that the location is inaccurate?
Ln227 lower buoyancy?
Ln253 “the embryos in the two epiphytic species of Bromheadia are encased in the seeds” - encased within the testa? Because, in my opinion, “the seed” includes the embryo, testa and other tissues (such as endosperm), if present
Ln258 wind speed is lower?
Ln259 higher buoyancy
Ln260 “great length and width seed” - of the seed.
Ln264 The sentence should be rephrased. It is an important consideration, to what extent the limited distribution range of certain species is related to seed dispersal, rather than suitable germination and/or growth conditions.
Ln271 perhaps, “in future research”?
Ln273 “absolutely” - fully? “in all the statements during” – in all stages of the preparation of the manuscript?
Reviewer 2 Report
The aim of the study was to reveal the differences between the seeds of one terrestrial and two epiphytic species in the Bromheadia genus, and confirm whether the species is compatible with the variability of the distribution range and life form along with the seed characteristics.
Although there are some inconsistencies with the qualitative and quantitative data of seeds and their volumes, it is an important finding to reveal information about the embryo structure, air spaces and morphology of seeds of species over long distances.
The presentation of the study and its scientific contribution is quite successful.
Kind regards.
Reviewer 3 Report
The manuscript entitled “Seed morphology, life form and distribution in three species of Bromheadia (Epidendroideae, Orchidaceae)” studies the differences between the seeds of one terrestrial species and two epiphytic in the genus Bromheadia to verify distribution range agreement with their seed variability and life forms. However, the number of studied taxa is low but as the manuscript gives a novel aspect of seed morphometry calculation it can go under further evaluation. Please see the following comments:
1- However, the manuscript is well-written but, in some parts, it needs moderate structural and grammatical language revision. Here are some examples:
L10: Seed morphology of three species of the genus Bromheadia → Seed morphology of three species belonged to the genus Bromheadia
L10: is analysed→ was analysed
L27: Testa cells showed a set of characters that have been observed under light and scanning electron microscope, providing qualitative and quantitative data: Please revise this sentence as the structure and verb tenses are ambiguous
L60: Bromheadia finlaysoniana: when you use the full scientific name for the first time use the abbreviated forms afterward.
L61: but there is lack of knowledge in the rest of the genus → but there is no information on the other members of the genus
2- The title could be changed as "Seed morphology, life form and distribution in three Bromheadia species (Epidendroideae, Orchidaceae)". This is up to the authors if they wanted to change the title.
3- The introduction is monotone and could be furnished with more information on the studied orchid species.
4- Please specify in material and method the exact origin of the studied species. Is there any geographical information on the collected seed? Is there any difference between exotypes regarding seed morphology? This would be valuable for population structure analyses.
5- As the embryo can be moved within the seed during imaging within the alcoholic solution or during coating (in case of SEM analysis), is it correct to evaluate the distance between the embryo and seed testa?
6- Please use smaller star symbols in Fig 5 A as the map and borders are hardly seen.
7- Regarding Fig 7, is there any statistical analysis to compare the mean? If the values are average it needs to bring the ±SE values as well.
8- You can cite the following reference (a study on seed morphometric study of terrestrial orchids) at the end of the first paragraph of the introduction along with [3-5] references:
Vafaee, Y., Mohammadi, G., Nazari, F., Fatahi, M., Kaki, A., Gholami, S., Ghorbani, A. and Khadivi, A., 2021. Phenotypic characterization and seed-micromorphology diversity of the threatened terrestrial orchids: implications for conservation. South African Journal of Botany, 137, pp.386-398.
Altogether I rate the manuscript as interesting and it needs major revision it could be considered for further evaluation after performing the required revisions.
Reviewer 4 Report
Dear editor the paper focused on valuable taxa that are faced to several threatened factors. However there are several shortcomings including:
Title
The title need to revise: ....distribution patterns of…
Introduction
The introduction play an important role to justify the study. There are some shortcomings including:
The Orchids are the most important taxa in conservation approach, so provide the suitable data on conservation biology and ecology of them
The authors should be write some data about orchids, ecology, biology distribution and etc
The hypothesis, questions and necessity should be described clearly.
The is not a suitable literature review to review history of study mainly about conservation management
The scientific names should be completed including author names
Study area
Where is the study area?
Main section in an ecological study is study area, so authors should describe study area in detailed including : Geology, geomorphology, climatology, pedology and ecology of target area.
Why light microscopy. As you know SEM showing better data on micro-morphology of plant structures.
Results
Are these differences intra or inter population?
A cluster or PCA analysis can be effective to represent differentiation or similarities.
Are there any correlation between ecology an morphological characters?
Distribution (3.2) is very weak
This section is very important
The authors should be extend the distribution patterns in detail including ecology, geo and etc
Discussion
The discussion is weak:
The authors should be justify the importance the study in conservation approach
Which data morphology provide to conservation ecology?
The discussion showing weakness in distribution patterns. Please complete it
The main achievements including conservation approach, taxonomy and biosystematics, ecology should be described clearly
The paper in current condition is not acceptable. The authors should be pay attention to comments. Authors should strongly promote the article. Accordingly after major revision must be assessed
Best Regards
Round 2
Reviewer 1 Report
The quality of the manuscript has improved substantially.
A monir check of language is required
Table 3: " seed volume (SV), embryo volume (EV), percentage of free air space" - it would be better to add the reference to the formulae that were used to calculate these values (making sure that these are not direct measurements)
Ln 331 "The decreasing of free air space" - decrease in (or reduction of)
Ln 336 " a great length" - a greater length?
Ln 351 " in which 351 grow the epiphytes" - at (?) which the epiphytes grow
Reviewer 3 Report
Thanks to the authors, they have done approximately all required revisions and it can be published in the present form. Just some minor issues:
1- The distribution map is usually placed before the other results and sometimes in material and methods.
2- The dendrogram in Fig 5 doesn't show the genetic relationship correctly as the main branch compacted at the end of the dendrogram. Could you please define the genetic distance scale of the dendrogram from 0-100 not 0-500?
There is no problem in other parts of the manuscript.
Reviewer 4 Report
Dear editor
the comments is corrected. Accordingly is acceptable
Author Response
Thank you very much for your comments.